# SYNERGISTIC CLASSIFICATION AND UNKNOWN DISCRIMINATION FOR OPEN SET RECOGNITION

## ABSTRACT

Deep learners tend to perform well when trained under the *closed set* assumption but struggle when deployed under *open set* conditions. This motivates the field of *Open Set Recognition* in which we seek to give deep learners the ability to recognize whether a data sample belongs to the known classes trained on or comes from the surrounding infinite world. Existing open set recognition methods typically rely upon a single function for the dual task of distinguishing between knowns and unknowns as well as making fine known class distinction. This dual process leaves performance on the table as the function is not specialized for either task. In this work, we introduce *Synergistic Classification and unknown Discrimination* (SCAD), where we instead learn specialized functions for both known/unknown discrimination and fine class distinction amongst the world of knowns. Our experiments and analysis demonstrate that SCAD handily outperforms modern methods in open set recognition when compared using AUROC scores and correct classification rate at various true positive rates.

## 1 INTRODUCTION

Recent studies have demonstrated the capacity of deep learners to achieve or even surpass human-level performance, particularly in the image recognition domain (He et al., 2015). This performance is typically achieved under the *closed set* assumption, however, in which the classes used for training the model are fixed and the model should only make predictions on this predefined set of classes. In practicality, the model may actually be deployed under *open set* conditions where the classes used for training are only a subset of the infinite surrounding world and the model must be able to distinguish between these known, trained on classes and the encompassing open world.

Conventionally, deep neural networks struggle under these open set conditions as they will confidently map unknown classes to the known class decision space (Nguyen et al., 2015; Hendrycks & Gimpel, 2017) as demonstrated in Figure 1a. This motivates the study of *Open Set Recognition* where we seek to discriminate between the world of *knowns* the model is trained on and the surrounding infinite *unknown* space.

Open set recognition was first formalized in Scheirer et al. (2013) and has since inspired an entire subfield of research. One of the first lines of work focused on an analysis of test time softmax scores (Hendrycks & Gimpel, 2017) as classifiers trained under the closed set assumption tend to produce low softmax probabilities for samples belonging to the unknown space. Bendale & Boult (2016) take a similar route by extending the softmax layer to allow prediction of an unknown class. These softmax based methods still suffer in open set recognition due to the inherent limitations of training the networks under the closed set assumption (Chen et al., 2020).

Other methods take a generative approach (Neal et al., 2018; Oza & Patel, 2019) in an attempt to generate samples belonging to the unknown world, or a distance-based approach (Mendes Júnior et al., 2017; Shu et al., 2020) by thresholding a distance to the nearest known class. While these methods perform better than traditionally used softmax score analysis, they still do not perform to their maximum capability as they have no true representation for what the world of unknowns may resemble.

Additionally, most current open set methods operate under the proposed setup of Scheirer et al. (2013) in which a single function is given the task of distinguishing between knowns and unknowns

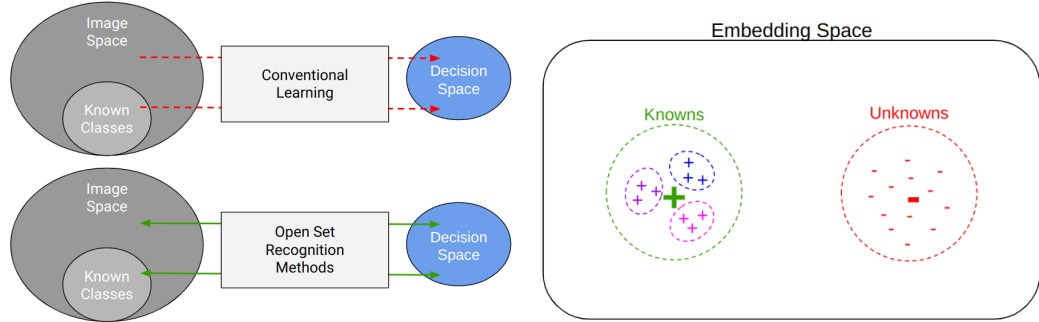

(a) In conventional learning, classes from the known space (training classes) and the infinite surrounding image space map to the same decision space. Open set recognition methods allow the decision space to map back to the infinite surrounding image space by means of an "unknown" label.

(b) In SCAD, we hypothesize that the embedding space should be distinctly separable between the known classes and unknown classes. The knowns and unknowns can be further separated by representing each cluster in the embedding space by their respective prototype depicted in bold above.

Figure 1: (a) The conventional closed set assumption and the proposed open set reocognition solution. (b) Our proposed hypothesis of embedding space separation between knowns and unknowns.

and additionally making fine distinction amongst the world of knowns (i.e, classification). This leads to a function that may perform relatively well for this joint task, but is not specialized for either task leaving performance on the table.

To this end, we introduce our method *Synergistic Classification and unknown Discrimination* (SCAD) to better address these shortcomings. In SCAD, we hypothesize that the known and unknown classes should clearly separate in the embedding space as demonstrated in Figure 1b. This separation can be accomplished by training an embedding network with a representative set of the unknown world referred to as *known unknowns* as in (Scheirer et al., 2014). Each embedding space can then be represented by its respective prototype for best separation. Furthermore, we train a classifier network under the closed set assumption for discrimination amongst the world of knowns. At test time, we can determine if a sample belongs to the world of knowns or unknowns by setting a threshold on the distance to the unknown prototype, and if a sample is deemed as known, we can query the classifier to determine its class. This formulation of two specialized decision functions allows each to be an expert in their respective task leading to higher performance when combined together.

## 2 RELATED WORK

**Open Set Recognition.** The field of open set recognition can be traced back to decision theory where we attempt to instill a classifier with a reject option when a classifier's confidence is low for a particular test sample (Bartlett & Wegkamp, 2008; Yuan & Wegkamp, 2010). Scheirer et al. (2013) first formalized the problem of open set recognition and explored the use of a "1-vs-all" SVM for unknown detection. Since then, deep learning methods have become the de facto method for open set recognition due to their great success. Bendale & Boult (2016) first introduce deep learning in the context of open set recognition by extending the softmax layer to modeling the distance of activation vectors based on extreme value theory. This approach was further extended by Ge et al. (2017) where deep neural networks are trained with unknown samples coming from a generative model. Other generative approaches include image reconstruction methods such as Oza & Patel (2019) and Yoshihashi et al. (2019) where unknown samples can be identified by poor reconstruction. More recently, prototype-based methods Chen et al. (2020); Shu et al. (2020); Chen et al. (2021) have shown great success by representing knowns and unknowns with learned prototypes and proceed to identify test samples based on distance to each prototype.

**Out-of-Distribution Detection.** Open set recognition is closely related to the field of out-of-distribution detection (Hendrycks & Gimpel, 2017) where we wish to identify if test samples come from a drastically different distribution. The key difference lies in open set methods' ability to fur-

ther distinguish fine labels amongst the world of knowns as mentioned in Boult et al. (2019). Liang et al. (2017) and Hsu et al. (2020) build upon the work of Hendrycks & Gimpel (2017) by performing post processing on the softmax confidence scores similar to the softmax method described above for open set recognition. Haroush et al. (2022) use hypothesis testing to generate $p$-values for each testing sample for determination of whether the sample comes from the in-distribution data. Zaeemzadeh et al. (2021) and Khalid et al. (2022) propose that learned features lie on a restricted low dimensional embedding space and the out-of-distribution data occupies the surrounding unrestricted space similar to the open set recognition methods Dhamija et al. (2018); Chen et al. (2020) and Chen et al. (2021). Our work draws inspiration from this described overlap region between open set recognition and out-of-distribution detection.

## 3 PRELIMINARIES

We first establish the formalities of the open set recognition problem before formulating our proposed solution (Scheirer et al., 2013; Geng et al., 2020; Chen et al., 2020). Suppose we are given a dataset $\mathcal{D}_{KK}$ of $n$ labeled data points we will refer to as *known knowns*, namely $\mathcal{D}_{KK} = \{(x_1, y_1), ..., (x_n, y_n)\}$ where $y_i \in \{1, ..., C\}$ is the label for $x_i$ for $C$ unique class labels in $\mathcal{D}_{KK}$. At test time, we will perform inference on the larger test data $\mathcal{D}_T$ consisting of data from $\mathcal{D}_{KK}$ as well as data from an unknown set $\mathcal{D}_{UU}$, which we refer to as *unknown unknowns*, whose labels $t_i \notin \{1, ..., C\}$. That is $\mathcal{D}_T = \mathcal{D}_{KK} \cup \mathcal{D}_{UU}$. We denote the embedding space of known category $k$ as $\mathcal{S}_k$ with corresponding open space $\mathcal{O}_k = \mathbb{R}^d - \mathcal{S}_k$ where $\mathbb{R}^d$ is the full embedding space consisting of known knowns and unknowns unknowns. We further define the positive open space from other known knowns as $\mathcal{O}_k^{pos}$ and the remaining infinite space consisting of unknown unknowns as the negative open space $\mathcal{O}_k^{neg}$, that is $\mathcal{O}_k = \mathcal{O}_k^{pos} \cup \mathcal{O}_k^{neg}$.

We first introduce open set recognition for a single known class and then extend to the multi-class scenario. Given the data $\mathcal{D}_{KK}$, let samples from known category $k$ be positive training data occupying space $\mathcal{S}_k$, samples from other known classes be negative training data occupying space $\mathcal{O}_k^{pos}$, and all other samples from $\mathbb{R}^d$ be unknown data, $\mathcal{D}_{UU}$, occupying space $\mathcal{O}_k^{neg}$. Let $\psi_k : \mathbb{R}^d \rightarrow \{0, 1\}$ be a binary measurable prediction function which maps the embedding $x$ to label $y$ with the label for the class of interest $k$ being 1. In this 1-class scenario, we wish to optimize the discriminant binary function $\psi_k$ by minimizing the expected error $\mathcal{R}_k$ as

$$\operatorname*{argmin}_{\psi_k}\{\mathcal{R}_k = \mathcal{R}_o(\psi_k, \mathcal{O}_k^{neg}) + \alpha\mathcal{R}_\epsilon(\psi_k, \mathcal{S}_k \cup \mathcal{O}_k^{pos})\} \tag{1}$$

where $\mathcal{R}_o$ is the open space risk function, $\mathcal{R}_\epsilon$ is the empirical classification risk on the known data, and $\alpha$ is a regularization parameter.

We can extend to the multiclass recognition problem by incorporating multiple binary classification tasks and summing the expected risk category by category as

$$\sum_{k=1}^{C}\mathcal{R}_o(\psi_k, \mathcal{O}_k^{neg}) + \alpha\sum_{k=1}^{C}\mathcal{R}_\epsilon(\psi_k, \mathcal{S}_k \cup \mathcal{O}_k^{pos}) \tag{2}$$

leading to the following formulation

$$\operatorname*{argmin}_{f \in \mathcal{H}}\{\mathcal{R}_o(f, \mathcal{D}_{UU}) + \alpha\mathcal{R}_\epsilon(f, \mathcal{D}_{KK})\} \tag{3}$$

where $f : \mathbb{R}^d \rightarrow \mathbb{N}$ is a measurable multiclass recognition function. From this, we can see that solving the open set recognition problem is equivalent to minimizing the combination of the empirical classification risk on the labeled known data $\mathcal{D}_{KK}$ and open space risk on the unknown data $\mathcal{D}_{UU}$ simultaneously over the space of allowable recognition functions $\mathcal{H}$.

## 4 METHODOLOGY

### 4.1 SYNERGISTIC CLASSIFICATION AND UNKNOWN DETECTION

In the traditional formulation of the open set recognition problem as described above, we assume a singular embedding space $\mathbb{R}^d$ consists of $N$ discriminant spaces for all known categories with all remaining space being the open space consisting of infinite unknowns. In formulating the framework

of SCAD, we instead postulate that the embedding space $\mathbb{R}^d$ is composed of two disjoint spaces, namely a known space $\mathcal{S}_{known}$ and an unknown space $\mathcal{O}_{unknown}$. That is to say that all of $\mathcal{D}_{KK}$ belongs to the space $\mathcal{S}_{known}$ and all of $\mathcal{D}_{UU}$ belongs to the infinite surrounding open space $\mathcal{O}_{unknown}$. Thus, the open space is formulated as $\mathcal{O}_{unknown} = \mathbb{R}^d - \mathcal{S}_{known}$

Under this new assumption of the embedding space, we can now pose a new formulation of the open set recognition problem by introducing a cascading optimization procedure where we wish to optimize both a binary prediction function $h : \mathbb{R}^d \to \{0, 1\}$ which maps the embedding of data $x$ to the label of known or unknown, and the classification function $f : x_i \to \mathbb{N}$ which maps the known data $x_i$ to their respective target label $y_i \in \{1, ..., N\}$ as

$$\underset{h}{\operatorname{argmin}} \left\{ \mathcal{R}_o(h, \mathbb{R}^d) \right\} \tag{4a}$$

$$\underset{f}{\operatorname{argmin}} \left\{ \mathcal{R}_\epsilon(f, \mathcal{S}_{known}) \right\} \tag{4b}$$

where $\mathcal{R}_o$ is the open space risk and $\mathcal{R}_\epsilon$ is the empirical classification risk. Based on this formulation we can see that the first optimization procedure leads to another binary prediction function $h$ similar to the traditional formulation while the second procedure leads to a multiclass prediction function $f$.

All that remains now is to find a method that best creates the full embedding space $\mathbb{R}^d$ to give a simple discriminant function $h$ and obtain a high performing multiclass prediction function $f$.

## 4.2 Embedding Separation of Knowns and Unknowns

We first focus on the discrimination between knowns and unknowns in the embedding space $\mathbb{R}^d$. A deep neural network $g_\theta : x \to \mathbb{R}^d$ is used as an embedding network to obtain embedding vectors for all data $x \in \mathcal{D}_{KK} \cup \mathcal{D}_{UU}$. In order to enforce the separation between the spaces $\mathcal{S}_{known}$ and $\mathcal{O}_{unknown}$, the triplet loss (Schroff et al., 2015) is a natural choice of loss function to use when training $g_\theta$. One could consider using other contrastive learning methods such as contrastive loss (Khosla et al., 2020) or tuplet loss (Sohn, 2016), however, the choice to use triplet loss was made as contrastive loss only considers pairs and tuplet loss is a more general version of triplet loss.

With the triplet loss, we can treat all training data in $\mathcal{D}_{KK}$ as the positive samples. For negative samples, we now need to find a representation of $\mathcal{D}_{UU}$ for modeling the space $\mathcal{O}_{unknown}$. Of course this open space and therefore this dataset is infinite, but we can use a representative set of $\mathcal{D}_{UU}$ we refer to as *known unknowns*, $\mathcal{D}_{KU} \subseteq \mathcal{D}_{UU}$, to train $g_\theta$ for embedding space separation of knowns and unknowns. The choice to use a representative training set $\mathcal{D}_{KU}$ to represent the entire world of unknowns is taken from out-of-distribution detection literature (Liang et al., 2017; Lee et al., 2018; Haroush et al., 2022).

Now armed with the known training set $\mathcal{D}_{KK}$ and representative unknown training set $\mathcal{D}_{KU}$, we can formalize use of the triplet loss to train $g_\theta$ as

$$\mathcal{L}_{g_\theta} = \sum_{i=1}^{n} ||g_\theta(x_i^a) - g_\theta(x_i^{KK})||_2^2 - ||g_\theta(x_i^a) - g_\theta(x_i^{KU})||_2^2 + \beta \tag{5}$$

where $x_i^a$ is a known known anchor, $x_i^{KK}$ is a known known positive sample, $x_i^{KU}$ is a known unknown negative sample, and $\beta$ is a margin that is enforced between the positive and negative pairs.

## 4.3 Discrimination Between Knowns and Unknowns

With a binary discriminant embedding space $\mathbb{R}^d$ now at hand, we must now develop the discriminant function $h$ to differentiate between knowns and unknowns. As such, we draw inspiration from Mensink et al. (2013); Ristin et al. (2014); Bendale & Boult (2016) by measuring the distance to the embedding prototypes for known/unknown discrimination. We represent each of the known and unknown clusters in the embedding space by their respective prototype determined by taking the means of the known knowns, $\mu_{KK}$, and known unknowns, $\mu_{KU}$, in the embedding space.

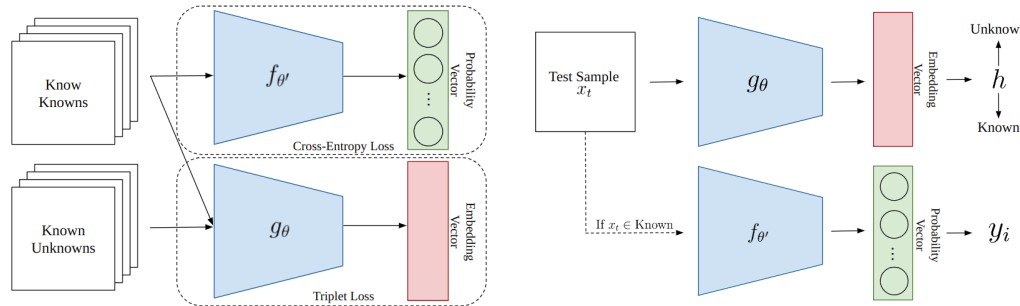

(a) We train a classifier network $f_{\theta'}$ using only data consisting of known knowns with cross-entropy loss and an embedding network $g_\theta$ using known knowns data and a representative set of the unknown data termed known unknowns with triplet loss.

(b) At test time we take a test sample and feed it to $g_\theta$ to get an embedding vector. We then feed that vector to the discriminator $h$ for known/unknown declaration. If known, we additionally feed the sample to the classifier $f_{\theta'}$ to obtain a fine class label.

Figure 2: SCAD training procedure (a) and inference procedure (b).

We then measure the Euclidean distance to $\mu_{KU}$ and set a threshold for final determination of whether a test sample is known or unknown. Thus, the binary function $h$ takes the form

$$h = \begin{cases} known & \text{if } d(g_\theta(x_t), \mu_{KU}) > \tau \\ unknown & \text{if } d(g_\theta(x_t), \mu_{KU}) \leq \tau \end{cases} \qquad (6)$$

where $x_t$ is a test sample from $\mathcal{D}_T$, $d(g_\theta(x_t), \mu_{KU}) = ||g_\theta(x_t) - \mu_{KU}||_2^2$ is the Euclidean distance between the embedding of $x_t$ and the known unknown prototype $\mu_{KU}$ and $\tau$ is a threshold.

## 4.4 MANAGEMENT OF OPEN SPACE RISK

In theory, the open space $\mathcal{O}_{unknown}$ is infinite making for difficult management of the open space risk $\mathcal{R}_o$. We instead opt to indirectly bound this open space for easier management of $\mathcal{R}_o$ as a direct bounding would be nearly impossible due to the infinite nature of $\mathcal{O}_{unknown}$. By enforcing the distance between samples from $\mathcal{S}_{known}$ and $\mathcal{O}_{unknown}$ to be outside some predefined margin of separation we are able to indirectly bound $\mathcal{O}_{unknown}$. This bounding procedure gives rise to Eq. 5 which enforces the distance between samples from the known knowns and known unknowns to be greater than or equal to the margin $\beta$.

The use of $\mathcal{D}_{KK}$ and $\mathcal{D}_{KU}$ in the training of $g_\theta$ for embedding space separation gives rise to the bounding spaces $\mathcal{B}_{known}$ and $\mathcal{B}_{unknown}$ respectively. Ideally, these spaces would be completely separable in $\mathbb{R}^d$, but in practicality there will be some overlap in the margin region. By representing each bounding space by its prototype as described above, we are able to achieve greater separation in $\mathbb{R}^d$. As a result, training with triplet loss for separation between $\mathcal{B}_{known}$ and $\mathcal{B}_{unknown}$ and further representing each bounding region with its appropriate prototype for final binary prediction can be viewed as managing the open space risk $\mathcal{R}_o(h, \mathbb{R}^d)$ in Eq. 4.

## 4.5 DISTINCTION AMONGST KNOWNS

The last remaining step is now developing a way to best identify which known class a sample belongs to for reduction of the empirical classification risk $\mathcal{R}_\epsilon$. In order to distinguish fine class labels amongst the world of knowns, we train a separate deep neural network $f_{\theta'}$ using cross-entropy loss in parallel with the embedding network $g_\theta$. As $f_{\theta'}$ is only concerned with classification of the knowns, we only use the data from $\mathcal{D}_{KK}$ to train the classifier. Figure 2a shows the full training procedure for training the multiclass prediction function $f_{\theta'}$ and the embedding network $g_\theta$.

At the inference stage, we only query $f_{\theta'}$ for a fine class label if the binary discriminant function $h$ predicts that a test sample $x_t$ belongs to the known space $\mathcal{S}_{known}$. Otherwise, $x_t$ is assigned to the world of unknowns. Figure 2b gives an overview for the entire inference stage.

## 5 EXPERIMENTS AND RESULTS

### 5.1 EXPERIMENTAL SETUP

**Datasets.** We test on four commonly used datasets in open set recognition literature. Each of the CIFAR datasets (Krizhevsky et al., 2009) is taken from either CIFAR10 or a combination of CIFAR10 and CIFAR100. For CIFAR10 experiments, all experiments are performed by treating the 6 non-vehicle classes as known classes and the remaining 4 vehicle classes as the unknown (i.e., open) classes. CIFAR+M experiments takes the 4 vehicle classes from CIFAR10 as known and randomly samples from M disjoint classes (i.e., non-vehicle classes) from the CIFAR100 dataset. Lastly, in Tiny-Imagenet experiments (Le & Yang, 2015) we randomly choose 20 classes as the known classes and treat all other 180 classes as unknown.

**Metrics.** We use the standard area under the ROC curve (AUROC) as the main metric when evaluating the performance of all compared methods. The benefit of using AUROC is its threshold independent measure of the binary open set discriminator and its ability to summarize each method's ability to distinguish between positive and negative instances across the various thresholds. A draw back of AUROC as commonly reported in open set trials, is it only takes into consideration known/unknown discrimination. A good open set recognizer should be able to additionally discriminate amongst the knowns given that a sample is predicted to be know. For this reason we additionally report the correct classification rate (CCR) at 95% true positive rate (TPR) of known detection similar to Dhamija et al. (2018).

**Compared Methods.** We compare our method, SCAD, to four open set recognition methods that are most comparable in regards to methodology. Counter-factual images (Neal et al., 2018) uses a GAN (Goodfellow et al., 2014) to generate counter examples to the known class which are then treated as the unknown class and used to train a "$K + 1$" classifier where the $(K + 1)^{th}$ class is the unknown class. Class anchor clustering (CAC) (Miller et al., 2021) poses a new loss function to entice each of the distinct known classes to cluster around their respective standard basis vector so that the unknown classes will then occupy the remaining open space. A distance threshold is then used for distinct known or unknown discrimination similar to SCAD. Adversarial Reciprocal Point Learning + confusion samples (ARPL+CS) (Chen et al., 2021) learns reciprocal points for each known class open space while simultaneously using a generator to generate confusing training samples to encourage known class separation in the latent space and uses a distance measure to the furthest reciprocal point to obtain a probability of belonging to a particular known class. Lastly, Vaze et al. (2022) propose that the best open set recognition model is simply one that is a Good Classifier for the closed-set scenario. With this good closed-set classifier at hand, an analysis of the maximum logit score produced by a sample is used in the final determination of distinct known or unknown.

**Setup.** For all methods, we train on the dataset splits described above. For neural network architectures, we use Resnet18 (He et al., 2016) in all tested methods for fairest comparisons except in counterfactual images and CAC. We keep the architectures unchanged in both of these methods as the former used a specific generator and discriminator for best GAN performance and the latter did not allow simplistic modulation with a Resnet encoder. Besides described architecture changes, all other hyperparemeters for compared methods remain unchanged. All methods are trained via SGD with standard L2 regularization. For SCAD, the margin of separation $\beta$ in Eq. 5 is set to 0.5 and a combination of semihard and hard negative mining are used for finding triplets. Lastly, we use half of unknown classes for all datasets as the training set $\mathcal{D}_{KU}$ in SCAD.

### 5.2 RESULTS COMPARISON.

We first evaluate the performance of SCAD vs. all other compared methods from an AUROC standpoint. Table 1 shows AUROC results averaged across 3 runs for all methods and Figure 3 shows the respective ROC curves. We observe that SCAD outperforms all compared methods for all datasets handily. This can be attributed to SCAD's specialized function $h$ for declaration of knowns and unknowns whereas all other methods use a singular function for both known/unknown discrimination and known class distinction as is commonly done in the traditional formulation of the open set recognition problem in Eq. 3.

Additionally, SCAD's $h$ discriminator is further assisted by clear known and unknown separation

Table 1: Reported AUROC score means and standard deviations for each tested method for the various tested datasets averaged over 3 runs.

| Method | CIFAR10 | CIFAR+10 | CIFAR+50 | Tiny-Imagenet |
|---|---|---|---|---|
| **Counter-Factual Images** | $0.6999 \pm 0.006$ | $0.8251 \pm 0.004$ | $0.8168 \pm 0.001$ | $0.5734 \pm 0.007$ |
| **Class Anchor Clustering** | $0.7156 \pm 0.002$ | $0.7425 \pm 0.013$ | $0.7721 \pm 0.002$ | $0.5452 \pm 0.036$ |
| **Good Classifier** | $0.7479 \pm 0.008$ | $0.7734 \pm 0.014$ | $0.7720 \pm 0.002$ | $0.6291 \pm 0.016$ |
| **ARPL+CS** | $0.7813 \pm 0.002$ | $0.8346 \pm 0.005$ | $0.8241 \pm 0.004$ | $0.6402 \pm 0.023$ |
| **SCAD (Ours)** | $\mathbf{0.9613 \pm 0.01}$ | $\mathbf{0.9223 \pm 0.023}$ | $\mathbf{0.9257 \pm 0.014}$ | $\mathbf{0.6548 \pm 0.0103}$ |

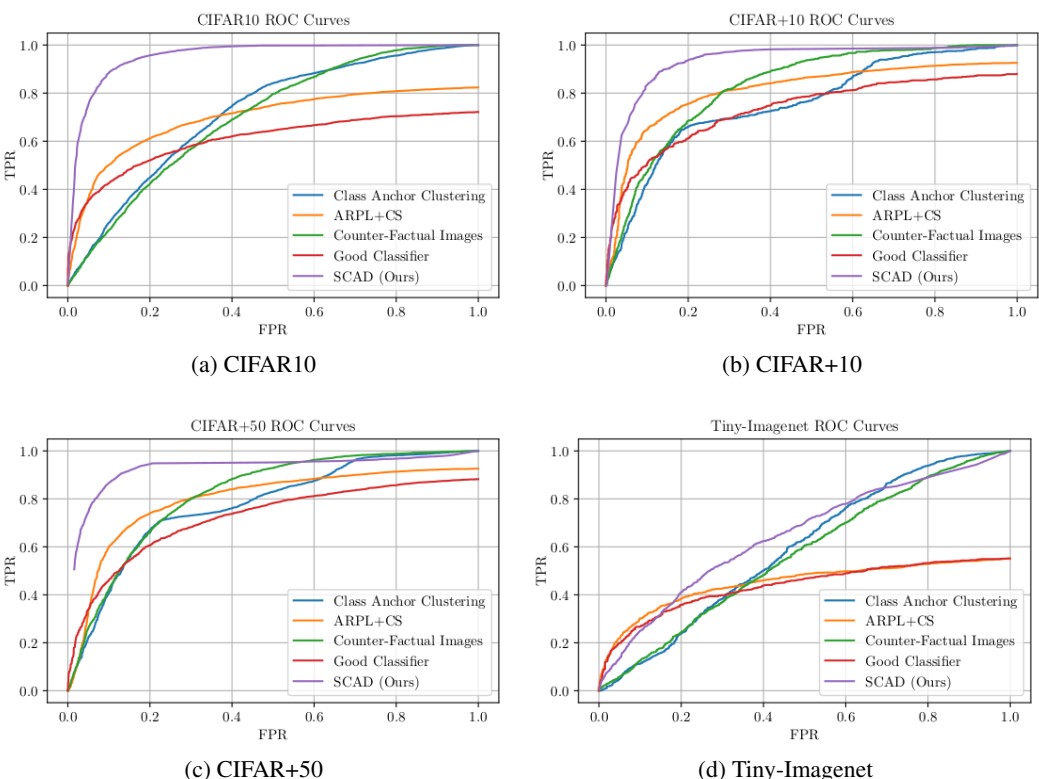

(a) CIFAR10        (b) CIFAR+10

(c) CIFAR+50        (d) Tiny-Imagenet

Figure 3: Corresponding ROC curves for each tested method for the various tested datasets.

in the embedding space $\mathbb{R}^d$ as initially hypothesized by means of the triplet loss. We can confirm this by analyzing the TSNE (Van der Maaten & Hinton, 2008) plot of the embeddings produced by $g_\theta$ as done in Figure 4 for the CIFAR10 data split. Of course, we observe an overlap region where discrimination between knowns and unknowns can prove challenging, but by representing each embedding cluster by its respective prototype, we are able to achieve better separation leading to a more favorable AUROC performance.

We do note the performance of SCAD vs. that of ARPL+CS and Good Classifier for Tiny-Imagenet in Figure 3d. While SCAD maintains a favorable AUROC score, there is a small region where these other two methods actually perform better. This would suggest in scenarios where small false positive rate (FPR) is desirable, one may want to consider alternatives to

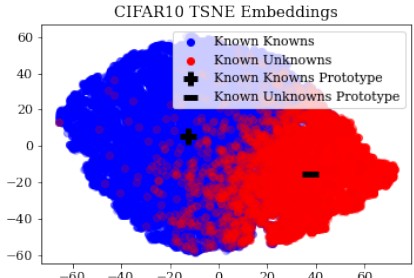

Figure 4: CIFAR10 TSNE plot of the embedding space.

SCAD. However, this small region of the ROC curve where SCAD is inferior is offset by the superior performance of SCAD in CCR elaborated on below.

Table 2: Reported CCR at 95% TPR score means and standard deviations for each tested method for the various tested datasets averaged over 3 runs.

| Method | CIFAR10 | CIFAR+10 | CIFAR+50 | Tiny-Imagenet |
|---|---|---|---|---|
| **Class Anchor Clustering** | $0.688 \pm 0.009$ | $\mathbf{0.8869 \pm 0.004}$ | $\mathbf{0.8805 \pm 0.007}$ | $0.3773 \pm 0.038$ |
| **Good Classifier** | $0.5650 \pm 0.001$ | $0.5731 \pm 0.012$ | $0.5694 \pm 0.003$ | $0.5263 \pm 0.002$ |
| **ARPL+CS** | $0.6571 \pm 0.002$ | $0.8233 \pm 0.002$ | $0.5821 \pm 0.004$ | $0.1732 \pm 0.004$ |
| **SCAD (Ours)** | $\mathbf{0.6962 \pm 0.004}$ | $0.8620 \pm 0.002$ | $0.8611 \pm 0.001$ | $\mathbf{0.6077 \pm 0.028}$ |

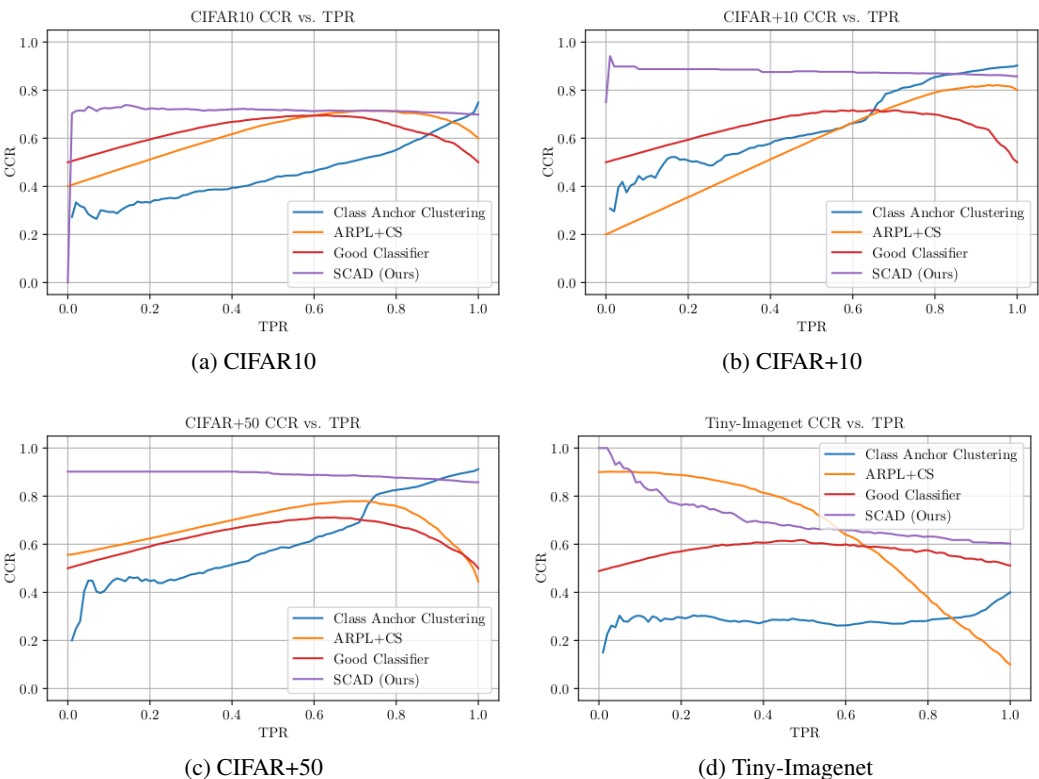

(a) CIFAR10

(b) CIFAR+10

(c) CIFAR+50

(d) Tiny-Imagenet

Figure 5: Corresponding CCR vs. TPR curves for each tested method for the various tested datasets.

We now evaluate the performance of SCAD against all other compared methods from a CCR standpoint. Table 2 reports the CCR at 95% TPR for all methods except Counter-Factual Images. We do not report results for Counter-Factual images due to the inherent nature of using a "$K + 1$" classifier (i.e., the "$K + 1$" classifier is not dependent on known/unknown discrimination as course distinction is based on discriminator scores and fine distinction amongst the "$K + 1$" classes is based on separate classifier scores). We overall observe that SCAD is mostly competitive with all other tested methods, but in particular performs exceptionally well on Tiny-Imagenet. The clear superiority of SCAD on Tiny-Imagenet can be attributed to having a specialized classifier $f'_\theta$ capable of making fine distinction amongst knowns for challenging datasets.

While SCAD remains competitive in all other datasets in regards to CCR at 95% TPR, we question if this is true for all operating TPRs. To answer this, we plot the CCR against various TPRs in Figure 5. From this, we make multiple interesting observations. Firstly, we can observe that SCAD is, in general, more stable than any of the compared methods. Again, this can be attributed to having a specialized classifier capable of consistent performance regardless of the number of known declarations. Secondly, we observe the CIFAR+10 and CIFAR+50 trials where SCAD is competitive, but not dominant in regards to CCR at 95% TPR. Figures 5b and 5c actually suggest that at nearly all other operating TPRs, SCAD is in fact superior. This would suggest that SCAD is the superior method in scenarios where higher TPRs can be waived.

We note the unintuitive performance of CCR being greater than 0 when TPR is 0. All methods except Good Classifier are distance based methods to some anchor point (e.g., distance to standard basis vector in CAC and distance to prototype in SCAD). Upon further inspection of these scenarios, few test samples are being correctly declared as known while the overwhelming majority are declared unknown. This can be attributed to a small amount of samples being infinitesimally close to their respective anchor allowing for correct declaration as known and thus leading to a non-trivial CCR at 0% TPR. This same principle applies to Good Classifier but in the context of logit scores.

## 5.3 PERFORMANCE ON KNOWN UNKNOWNS VS. UNKNOWN UNKNOWNS

We now turn our attention to analyzing the impact of using a representative set of the unknowns, $\mathcal{D}_{KU}$, when training the embedding space $\mathbb{R}^d$ and how this might generalize to the entire world of unknowns, $\mathcal{D}_{UU}$. To do so, we partition the testing data into two disjoint testing sets with resepct to the unknown data: one testing set contains only known unknowns while the other contains only unknown unknowns. We report the AUROC for each of these testing sets in Table 3 averaged over 3 runs.

Table 3: Reported AUROC score means and standard deviations for each disjoint unknown test set for the various tested datasets averaged over 3 runs.

| Unknown Dataset | CIFAR10 | CIFAR+10 | CIFAR+50 | Tiny-Imagenet |
|:---:|:---:|:---:|:---:|:---:|
| $\mathcal{D}_{KU}$ | $0.970 \pm 0.001$ | $0.925 \pm 0.019$ | $0.952 \pm 0.006$ | $0.640 \pm 0.037$ |
| $\mathcal{D}_{UU}$ | $0.9347 \pm 0.024$ | $0.8712 \pm 0.001$ | $0.944 \pm 0.005$ | $0.6269 \pm 0.033$ |

We observe that the difference in performance between $\mathcal{D}_{KU}$ and $\mathcal{D}_{UU}$ is relatively small. Even the isolated performance of $\mathcal{D}_{UU}$ still outperforms all other compared methods in Table 1 suggesting the the representative set $\mathcal{D}_{KU}$ allows the embedding model $g_\theta$ to generalize well to the world of unknowns. Furthermore, we note the small disparity in AUROC scores for each of the unknown datasets in the CIFAR+50 and Tiny-Imagenet trials compared to that of CIFAR10 and CIFAR+10. Since we are using half of the entirety of unknown classes as the representative set $\mathcal{D}_{KU}$ in SCAD, this suggests that the larger we can make the representative training set, the better our ability to generalize to the entire world of unknowns will be.

## 6 CONCLUSION

In this work, we introduce our method SCAD for open set recognition. SCAD benefits from having two specialized functions for known and unknown discrimination as well as fine class distinction amongst knowns. This allows each function to be an expert for their respective task allowing for top tier performance compared to that of traditional open set recognition methods where a single function is used for both known/unknown discrimination and fine class distinction. Additionally, by using a representative set of the unknowns termed *known unknowns*, we are able to train an embedding network for distinct separation between knowns and unknowns in the embedding space allowing for easy discrimination. Our experiments show that we outperform modern open set recognition methods in not only known/unknown discrimination, but also correct classification amongst the knowns.

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
