# OpenReview forum: "Synergistic Classification and Unknown Discrimination for Open Set Recognition"
_ICLR.cc/2024/Conference — ICLR 2024 Conference Withdrawn Submission_

### Official Review · Reviewer_oZJx · 2023-10-31

**Soundness:** 3 good
**Presentation:** 3 good
**Contribution:** 2 fair
**Rating:** 5
**Confidence:** 5

**Summary:**

This submission proposes a new approach to open-set recognition.
Existing OSR methods reuse classification networks' embeddings (class activations)
with threshold-based or distance-based unknown detection without proving that it is an optimal approach.
Then, this work proposes Synergistic Classification and unknown Discrimination
(SCAD). Exploiting "known unknowns" as a part of training data,
SCAD learns a feature extractor specialized for unknown detection.
Experiments with CIFAR-10/50 and Tiny ImageNet show
that SCAD outperformed existing OSR methods
(but in my understanding, that are not exposed to known unknowns in training.) with margins.

**Strengths:**

- Separating networks for classification and unknown detection is a not-deeply investigated and
 potentially useful idea.

- Using known unknowns in training would be a practical direction, although existing researches avoided it (seemingly to keep theoretical conciseness?).

**Weaknesses:**

- Comparisons with existing methods that do not use known unknown seem unfair.
All compared methods were trained without exposure to known unknowns during training.
Thus, only outperforming them does not reveal which is the source of improvement,
design of SCAD or usage of known unknowns in training.
I think comparing known-unknown-free version of SCAD or existing methods + known-unknown training
is needed to validate the proposed method.

- Sensitivity to selection of known unknown classes is not investigated.
I could not find how known unknowns and unknown unknowns are separated for experiments.
Intuitively, SCAD may be strong when known unknowns are luckily similar to unknown unknowns
but may not be so strong when unknown unknowns come from distant classes from known unknowns.

- Using known unknowns in training is analogous to outlier exposure in anomaly detection.
Recommended citation:
[a] Deep Anomaly Detection with Outlier Exposure, ICLR2019

**Questions:**

na

---

### Official Review · Reviewer_DxdR · 2023-10-31

**Soundness:** 1 poor
**Presentation:** 2 fair
**Contribution:** 1 poor
**Rating:** 3
**Confidence:** 3

**Summary:**

This paper is another example of using an extra binary classifier to address ood task. They construct the binary classifier with a metric learning approach but the novelty is limited, and experiments are not sufficient.

**Strengths:**

The paper is clearly and well written.

**Weaknesses:**

1) The proposed method lacks novelty. The idea of constructing or using feature space not utilized by known classes for open class is not new. And the intuition using metric learning to make known class more compact for easier ood using feature distance is not novel either. In a non-ood paper, Large-Scale Long-Tailed Recognition in an Open World, the authors used center loss to make known classes more compact and use feature distance to for ood. And it was almost 5 years ago. Simply changing center loss to multiple binary triplet loss is not sufficient.
2) More importantly, none of the newer ood methods, for example One-Ring: A Simple Method for Source-free Open-partial Domain Adaptation, are fully discussed or compared in the experiments. Therefore, the robustness of the proposed method is not justified, In addition, One-Ring, even though uses additional classification dimension, also follows the intuition of utilizing non-occupied space for ood.

**Questions:**

As above.

---

### Official Review · Reviewer_32wB · 2023-10-31

**Soundness:** 2 fair
**Presentation:** 3 good
**Contribution:** 2 fair
**Rating:** 3
**Confidence:** 5

**Summary:**

This paper introduces a novel approach called Synergistic Classification and Unknown Discrimination (SCAD). It enhances Open Set Recognition (OSR) by concurrently incorporating samples from both known and unknown classes during the training phase. The experimental results demonstrate the effectiveness of this method.

**Strengths:**

The paper is presented clearly and easy to follow.

**Weaknesses:**

(1)	Some important related works are missing.
The author proposes to use known unknowns (i.e., outliers) during the training phase to constrain the region for unknowns, and this idea has been explored in many prior works, with one of the most representative being Outlier Exposure (OE) [Hendrycks et al. 2019]. OE achieves remarkable performance by solely requiring a single network to concurrently manage classification and the rejection of unknowns. It's worth noting that this paper doesn't make any references to OE, and there is a noticeable lack of in-depth discussion or comparison concerning methods and experimental results. Essentially, the primary contributions introduced in this paper have already been covered in OE.

(2)	The choice of known unknowns is unfair.
In the original paper of OE, for fair comparison, the authors ensured that the known unknowns (i.e., outliers) collected and the unknowns in the testing phase had no overlapping classes. They carefully removed those unknowns with the same categories. However, in the setup in Section 5.1 of this paper, the authors employed half of the unknown classes for all datasets, making the comparison unfair to implement.

(3)	Lack of an important baseline recently proposed.
The author has mentioned that the recent work [Vaze et al., 2022] finds that simply training a network on the closed set can achieve the SOTA performance. However, the author does not provide a detailed comparison with this baseline in the paper. Moreover, despite the use of a slightly different network architecture, the results of (ARPL+CS) in this paper are significantly lower than those in [Vaze et al., 2022], which compromises the reliability of the experimental outcomes. To enhance the credibility of the findings, the author should furnish more detailed results.

References:
[Hendrycks et al., 2019] Deep Anomaly Detection with Outlier Exposure. ICLR 2019.
[Vaze et al., 2022] Open-Set Recognition: a Good Closed-Set Classifier is All You Need? ICLR 2022.

**Questions:**

Please refer to [Weaknesses]. The main concerns are about the contributions and experimental setups.

---

### Official Review · Reviewer_RKPT · 2023-11-03

**Soundness:** 3 good
**Presentation:** 3 good
**Contribution:** 2 fair
**Rating:** 3
**Confidence:** 4

**Summary:**

Deep learning models excel in recognizing images they've been trained on but falter with images outside their training set, known as open set recognition (OSR). Traditional methods, which adjust softmax layers or use generative and distance-based approaches, are limited because they don't effectively represent or separate unknown classes. Authors proposed SCAD method tackles this by learning two specialized functions: one to distinguish between known and unknown classes and another for classifying known classes. SCAD trains with "known unknowns" to enhance the distinction between classes in the embedding space and uses prototypes to set discrimination thresholds. This specialized dual-function approach allows SCAD to outperform existing methods in OSR, as evidenced by higher AUROC scores and correct classification rates.

**Strengths:**

The task of open set recognition is of interest to the field.

**Weaknesses:**

1. The presentation of the paper can be confusing sometime. E.g. in Fig 2., please explain in the caption the 'know knowns' and 'known unknowns', which are very confusing. This makes understand the core idea inaccessible.
2. Large-scale experiments like original Imagenet and iNaturalist is needed. Current experiments are too small-scale to verify the proposal.
3. some ablation studies are missing. E.g. \beta in the Eqn. 5.

**Questions:**

1. Fig.1b hypothesize that the embedding space should be distinctly separable between the known classes and unknown classes. However, in Fig 4, I did not see distinc separation between the two. I also do not think the hypothesis holds in general. Can authors expalin.